# Impact of Vaccination on the Course and Outcome of COVID-19 in Patients with Multimorbidity

**DOI:** 10.3390/vaccines11111696

**Published:** 2023-11-07

**Authors:** Kirill Lomonosov, Alyona Lomonosova, Alla Mindlina, Roman Polibin, Maksim Antipov, Gleb Grimm

**Affiliations:** Federal State Autonomous Educational Institution of Higher Education I.M. Sechenov First Moscow State Medical University of the Ministry of Health of the Russian Federation (Sechenov University), Moscow 119991, Russia; lomonosov_k_s@student.sechenov.ru (K.L.); mindlina_a_ya@staff.sechenov.ru (A.M.); polibin_r_v@staff.sechenov.ru (R.P.); antipov_m_o@staff.sechenov.ru (M.A.); grimm_g_v@staff.sechenov.ru (G.G.)

**Keywords:** COVID-19, multimorbidity, epidemiology, vaccination

## Abstract

Vaccination is the most cost-effective method of preventing COVID-19; however, data on its effect on patients with multimorbidity is limited. The aim was to evaluate the effect of vaccination against new coronavirus infection (NCI) in patients with multimorbid pathology in hospital treatment on the outcome of COVID-19 disease. An analysis was carried out of 1832 records of patients in one of the COVID-19 hospitals in Moscow for 2020–2022. Statistical analysis was carried out using the StatTech v. 3.1.3 software, and the binary logistic regression (BLR) method was used to obtain prognostic models. The median age of patients was 69 years, and 76% of them had received two vaccine doses. To assess the outcome of the disease, two prognostic models were obtained depending on the presence of a multimorbidity in patients: cardiovascular pathology and/or atherosclerosis and/or type 2 diabetes mellitus (Model 1) or atherosclerosis and/or type 2 diabetes mellitus and/or encephalopathy (Model 2), against the background of the presence or absence of vaccination against NCI. When assessing the outcome of NCI in Model 1, the odds of death decreased by 3.228 times with two doses of Sputnik V in patients with multimorbidity. According to Model 2, for patients with multimorbidity, the chances of death decreased by 3.281 times with two doses of Sputnik V. The presence of two doses of Sputnik V increased the likelihood of recovery in patients with multimorbidity by more than three times.

## 1. Introduction

The SARS-CoV-2 infection (COVID-19) remains an urgent problem for modern healthcare [1,2]. COVID-19 is a trigger factor for the development of many pathological conditions in the pulmonary, cardiovascular, endocrine, and immune systems, while also potentially leading to the development of pneumonia, respiratory failure, and death [3].

At the same time, according to the results of a significant number of studies on the pathogenesis of COVID-19, factors that significantly increase the likelihood of a lethal outcome of the disease are concomitant chronic diseases in a patient, and multimorbidity is a leading risk factor associated with a more severe outcome after SARS-CoV-2 infection [4]. Multimorbidity itself is one of the greatest challenges facing health services worldwide [5]. When it comes to estimating the prevalence of multimorbidity by country, several large systematic reviews show heterogeneous results across studies. Among the total population over 18–19 years of age, the top five countries in terms of multimorbidity included the USA (54.1%), Germany (39.6%), Denmark (37%), Korea (34.8%), and Canada (33.5%). It should be noted that such estimates may vary depending on the approaches used by the authors. The prevalence rate varied significantly in studies where multimorbidity was established based on medical records and patient questionnaires, and a high correlation with the age and gender of persons included in the study was shown. For example, in the prevalence of multimorbid conditions among people over 50 years of age, countries such as the United Kingdom (80.8%), Ireland (73.3%), the USA (72.4%), and the Russian Federation (70%) were the most prominent. Among patients over 60 years of age, the leading positions are occupied by China (74.3%), India (79.4%), and Brazil (92.5%), in which, for example, among people over 50 years of age, comorbidity occurs in 67.8% cases and only 29.1% cases among the total population over 20 years of age. Multimorbidity was significantly more common in women (39.4%, 95% CI = 36.4–42.4%) than in men (32.8%, 95% CI = 30.0–35.6%) [6]. Thus, it has been shown that on average, every third patient is in a state of multimorbidity and belongs to a high-risk group for an unfavorable outcome of COVID-19, and therefore it is important to study the impact of preventive measures on this group of patients and develop new approaches and programs to protect them. However, most studies to date have focused on pre-existing single chronic conditions and not complex multimorbid combinations.

As is known, vaccination is one of the most cost-efficient and effective methods of preventing infectious diseases in the modern world, and COVID-19 is no exception [7,8]. Numerous studies in Russia and other countries have shown that vaccination against SARS-CoV-2 has a beneficial effect on the course of the disease in patients with a negative pre-morbid medical history; however, most studies have assessed the effect of vaccination in a group of patients with a single condition. For example, an analysis of medical observation data in the United States, including 10,162,227 patients, showed that in the group of patients with cardiovascular diseases, the presence of vaccination was associated with a lower incidence of complications and a severe course of coronavirus disease [9]. In the UK, a population-wide cohort study conducted from December 2020 to March 2021 found that patients vaccinated with Pfizer/BioNTech (BNT162b2) and AstraZeneca had a lower risk of developing myocardial infarction even 28 days after vaccination [10]. A population-based study conducted in South Korea showed a 52% reduction in the risk of acute myocardial infarction and ischemic stroke after COVID-19 in fully vaccinated patients (two vaccinations, 28 days later) [11].

According to a prospective study conducted in Russia, which included 1070 patients with confirmed coronavirus disease, it was shown that the incidence of thrombosis of various localizations, including PATE (pulmonary artery thromboembolia) and deep leg vein thrombosis, was statistically higher in the group of unvaccinated individuals compared with the group of patients vaccinated with Sputnik V [12].

The efficacy of coronavirus vaccination has also been evaluated in individuals with one or more chronic noncommunicable diseases (CNCDs), such as cardiovascular disease, diabetes mellitus (DM), chronic obstructive pulmonary disease (COPD), cancer, and immunosuppressive conditions. A retrospective analysis of 800 patients aged 18 to 90 years who were vaccinated (Sputnik V (Gam-Covid-Vac, developed by the Gamaleya Research Institute of Epidemiology and Microbiology, Moscow, Russia)) from February to May 2021 was carried out, and serological monitoring was carried out at 21 and 42 days after vaccination. The study showed that the level of IgG in all groups increased fourfold at day 42 compared with the visit at day 21, which may indicate the immunological efficacy of vaccination for people with CNCDs [13].

A recent literature review presents data on the use of various vaccines in people with diabetes mellitus (DM) and obesity, showing the high efficacy of immunization in these groups. Experience with COVID-19 vaccination using various vaccines (Moderna mRNA-1273, Pfizer-BioNTech, BNT162b2, AstraZeneca COVID-19 vaccine AZD1222, SII Covishield, SK Bioscience, and Sputnik V) showed similar safety and efficacy profiles among patients from risk groups such as those with obesity and diabetes [14].

Today, the risks of COVID-19 remain high. Since COVID-19 can vary in severity—from asymptomatic forms to severe, extremely severe, and fatal issues related to the search for predictors of a severe course and an unfavorable outcome of a coronavirus infection—remain extremely important today. At the same time, we found only a limited number of studies that studied the effect of multimorbidity on the outcome of COVID-19 and the effect of vaccination on the survival of patients with multimorbid conditions [15]. In this regard, our study of the effect of vaccination on the course and outcome of COVID-19 in patients with multimorbidity undergoing in-patient treatment in Moscow is extremely relevant.

Our study aimed to establish the effect of vaccination against a new coronavirus infection in patients with multimorbidity in hospital treatment on the severity, duration, and outcome of COVID-19.

## 2. Materials and Methods

We analyzed 1832 case histories of patients from one of the temporary COVID-19 hospitals in Moscow, hospitalized in 2020–2022. Statistical analysis was carried out using StatTech v. 3.1.3 (developer: StatTech LLC, Kazan, Russia). Figures and tables were created using the Microsoft Office Excel 2016 software. Quantitative indicators were assessed for normality using the Shapiro–Wilk test (with the number of subjects < 50) or the Kolmogorov–Smirnov criterion (with the number of subjects > 50). In the absence of a normal distribution, quantitative data were described using the median (Me) and the lower and upper quartiles (Q1–Q3). Categorical data were described with absolute values and percentages. A comparison of two groups with a normal distribution was performed using the Student’s t-test. A comparison of two groups in terms of a quantitative indicator, the distribution of which differed from the normal one, was performed using the Mann–Whitney U-test. Comparison of three or more groups in terms of a quantitative indicator, the distribution of which differed from the normal one, was performed using the Kruskal–Wallis test, and post hoc comparisons were performed using Dunn’s test with Holm’s correction. A comparison of percentages in the analysis of four field and multi-field contingency tables was performed using Pearson’s chi-square test (with expected phenomenon values of more than 10). A predictive model characterizing the dependence of a quantitative variable on factors was developed using the linear regression method. The construction of a predictive model of the outcome probability was carried out using logistic regression. Nigelkirk’s R^2^ coefficient served as a measure of certainty, indicating that part of the variance can be explained using logistic regression.

## 3. Results

When assessing the epidemiological situation of the new coronavirus infection in the Russian Federation during the period of collecting data from patients included in our study, according to the portal “Stopcoronavirus.RF” at the beginning of 2022, 11,000,598 cases of COVID-19 were registered in the Russian Federation, and 321,717 cases were fatal.

According to Rosstat (Federal State Statistics Service), at the beginning of 2022, 145,478,097 permanent residents were registered in Russia, of which 73,402,606 people were vaccinated (166,184,558 administered doses), which corresponds to 51% vaccination coverage. According to the results of the assessment, at the time of the statistical analysis, the percentage of those who had contracted COVID-19 was 12.5%, of which 96.6% recovered.

Moscow accounted for 15.1% of all cases of COVID-19 in the Russian Federation; the increase in the incidence of COVID-19 in 2021 was 34.2% and 3.8% in 2022 (1,220,975 and 1,269,659 cases, respectively) (Table 1). The reasons for that may include better diagnostic capabilities as well as the fact that Moscow is a metropolis with a permanent population of over 13 million people. The Moscow agglomeration is the largest in Europe and one of the largest agglomerations in the world. In the conditions of a megacity, many risk factors are realized, resulting in a high level of non-communicable diseases in the population as well as an increase in multimorbidity, which indicates the importance of protecting the population from additional threats, such as infectious diseases, including COVID-19. All the above indicates the high relevance of conducting the study in Moscow.

Since in a metropolis there is a high prevalence of multimorbid conditions among the population, which are the leading risk factors for severe disease, and vaccination is the most effective method of combating COVID-19, we decided to conduct a study to assess the effectiveness of immunization against COVID-19 and the degree of protection against a severe course of the disease and mortality among patients with multimorbidity.

Analyzing the structure of the database we collected, we found that, based on information from 1832 patients in one of the temporary COVID-19 hospitals in Moscow hospitalized in 2020–2022, the median patient age was 69 years (Q1–Q3 = 57–79, min 18 years, max 99 years), and 40.7% of patients were male (n = 745) (Figure 1). A total of 283 patients (15.4%) were employed, and 1549 (84.6%) patients were not officially employed at the time of hospitalization (Figure 2).

After analyzing the data, we found that among hospitalized patients, 85.6% (1568 patients) did not receive even one dose of vaccine, while gender and age did not affect the availability of vaccination (*p* = 0.15 and *p* = 0.295, respectively). However, there were significant differences in adherence to vaccination among the employed and unemployed (Table 2). The odds of being vaccinated in the group of non-working patients were 1.547 times lower compared to the group of officially employed patients; the difference in odds was statistically significant (OR = 0.647; 95% CI: 0.466–0.898).

When analyzing the vaccination status by number of doses, we found that almost 76% of patients had completed the two dose vaccination course (155 people were vaccinated twice with Sputnik V, twenty-six patients had two doses of CoviVac and nineteen patients had two doses of EpiVacCorona), and 5.7% of patients had a history of two vaccinations and one revaccination with Sputnik Light. Among patients vaccinated with a single dose (18.6%), forty-two people were vaccinated with Sputnik V/Light, four received an injection of CoviVac, and three received EpiVacCorona (Figure 3).

To assess the impact of vaccination on COVID-19 outcomes, we selected four endpoints: the probability of the pathogen being released into the environment (assessed by the result of SARS-CoV-2 PCR testing); the severity of lung damage (assessed by CT); the length of hospital stay; and the outcome of the disease (death or recovery). Data on the distribution of patients by selected groups are presented in Figure 4, Figure 5, Figure 6 and Figure 7.

Next, we evaluated each of the parameters to assess the possibility of including each of them in the regression model. According to statistical calculations, it was found that the presence of vaccination did not affect the result of the PCR test (*p* = 0.647), while at the time of admission, the PCR test was positive in 87.4% of patients. According to the results of CT, only 1% did not reveal lung damage, almost half of the patients (44.9%) were diagnosed with the CT2 stage, a third of patients had first-degree lung damage, and CT3 and CT4 stages were diagnosed in 18.8% and 5.5% of patients, respectively. When assessing the impact of previous vaccination on the degree of CT in the disease, significant differences were found between groups of vaccinated and unvaccinated patients (*p* = 0.002). In the group of individuals with CT1–4 stage damage, vaccinated persons were 3.738 times less common than in the group without lung injury (OR = 0.268; 95% CI: 0.110–0.652).

To assess the impact of vaccination on hospital stay duration, we developed a predictive model using binary logistic regression. The number of observations was 1812 since 20 patients had no data on the duration of treatment. The resulting regression model was statistically significant (*p* < 0.001). It shows that if vaccination was available, the chance that the patient would be hospitalized for more than 2 weeks decreased by 1.816 times. Next, we decided to determine which of the vaccines had the most impact on protecting patients (Table 3). According to the calculation results, the most significant result was obtained in the group of patients immunized twice with the Sputnik V vaccine and in the group of hospitalized, unvaccinated patients. This may be explained by the relatively smaller number of patients immunized with vaccines from other manufacturers and/or using different vaccinations. In this regard, we included only these two groups for further analysis: unvaccinated individuals and patients with two doses of Sputnik V.

When studying the effect of vaccination on the outcome of the disease in patients, we excluded 49 patients from the statistical analysis since they were transferred to another medical organization and their disease outcome could not be determined (in total, 1783 patients were included in the calculation of the effect of vaccination on the outcome). According to the data obtained, when comparing groups with and without previous vaccination, statistically significant differences were established depending on the outcome of the disease (*p* = 0.025). The chances of recovery in the vaccinated group were 1.975 times higher than in the unvaccinated group; the difference in odds was statistically significant (95% CI: 1.077–3.620).

Therefore, vaccination against COVID-19 significantly reduces the adverse course of the disease. However, despite such significant results, doctors frequently give unreasonable medical exemptions from vaccination to patients with chronic diseases and/or multimorbidity. Therefore, one of the objectives of our study was to show that the impact of vaccination was the most significant for this particular group of patients. To do this, we developed a predictive model using binary logistic regression to determine the probability of recovery or death from COVID-19, depending on the number of vaccine doses administered previously while taking into account the age and gender of patients. Based on the results of the evaluation of parameters carried out at the first stages of the study, to ensure the reliability of the analysis, we only took two groups of patients into the study: unvaccinated patients and those vaccinated with two doses of Sputnik V (n = 1677). The endpoint was the patients’ disease outcomes (death or recovery).

We developed a predictive model to determine the likelihood of a favorable COVID-19 outcome, depending on the presence of a multimorbid condition in patients (cardiovascular disease and/or atherosclerosis and/or type 2 diabetes) and the presence or absence of vaccination against COVID-19 using binary logistic regression. The number of observations was 1677. The observed connection can be described by the following equation:P = 1/(1 + e^−z^) × 100%,
Z = −3.150 + 0.471X_one condition_ + 1.527X_two conditions_ + 2.241X_three conditions_ − 1.172X_Sputnik_,
where P—chance of death, X_one condition_ − CVD + Atherosclerosis + Diabetes (0—no disease, 1—one of the listed conditions), X_two conditions_ − CVD + Atherosclerosis + Diabetes (0—no disease, 1—two of the listed conditions), X_three conditions_ − CVD + Atherosclerosis + Diabetes (0—no disease, 1—three of the listed conditions), and X_Sputnik_ − Vaccination (0—not vaccinated, 1—Sputnik)

The final regression model is statistically significant (*p* < 0.001). Based on the value of the Nigelkirk R^2^ coefficient of determination, the model explains 9.2% of the observed variance in the indicator characterizing the outcome of disease in a patient.

When assessing the probability of a favorable outcome of COVID-19 depending on the presence of multimorbidity in patients (cardiovascular disease and/or atherosclerosis and/or type 2 diabetes mellitus), the chances of death increased by 4.604 times when there were two of the listed diseases and by 9.402 times when there were three. When assessing the impact of vaccination in this group, the chances of death were reduced by 3.228 times with two doses of Sputnik V (Table 4) (Figure 8).

We also obtained a reliable predictive model for determining the probability of a lethal outcome depending on vaccination and diseases such as atherosclerosis and/or type 2 diabetes and/or encephalopathy in patients by binary logistic regression. The observed connection can be described by the following equation:P = 1/(1 + e^−z^) × 100%, 
Z = −2.950 − 1.188X_Sputnik_ + 0.488X_Diabetes_ + 1.477X_Atherosclerosis_ + 0.824X_Encephalopathy_,
where P is the probability of death; X_Sputnik_ − vaccination (0—not vaccinated, 1—Sputnik); X_Diabetes_ − Type 2 Diabetes (0—no diabetes, 1—diabetes); X_Atherosclerosis_ − Atherosclerosis (0—no atherosclerosis, 1—atherosclerosis); and X_Encephalopathy_ − Encephalopathy (0—no encephalopathy, 1—encephalopathy)

The final regression model is statistically significant (*p* < 0.001). Based on the value of the Nigelkirk R^2^ coefficient of determination, the model explains 13.1% of the observed variance in the indicator characterizing the outcome of disease in a patient.

If the patient had a history of vaccination, the chances of death were reduced by 3.281 times with two doses of Sputnik V. As for chronic diseases, the chances of death increased by 1.629 times in the presence of diabetes, by 4.378 times in the presence of atherosclerosis, and by 2.279 times with encephalopathy (Table 5) (Figure 9).

Thus, we found that vaccine prevention of COVID-19 in the adult population using Russian-made vaccines for patients with multimorbidity has high epidemiological effectiveness.

## 4. Discussion

Considering both the rapidly growing prevalence of the population who have had SARS-CoV-2 infections and the growing number of people living with multimorbidity around the world (an average of 5.8 diseases per person), determining the nature and impact of vaccination in patients with multimorbidity (defined as two or more conditions) on the risk of severe SARS-CoV-2 infection has important clinical and public health implications that are not captured by focusing on individual diseases [16]. In high- and lower-middle-income countries (HICS and LMICS), the prevalence of multimorbidity was 37.9% (95% CI: 32.5–43.4%) and 29.7% (26.4–33.0%), respectively [17]. In Russia, more than 44% of the adult population of the country meets similar criteria [18].

The health burden of patients with multimorbidity is significant and is linked with increased mortality. A recent meta-analysis of 5806 studies concerning the link between multimorbidity and mortality (26 studies included) demonstrated a risk ratio of 1.73 (95%CI: 1.41, 2.13) and 2.72 (95%CI: 1.81, 4.08) for people with two or more and three or more diseases, respectively [19]. Given this, COVID-19 infection poses a serious danger and can increase the likelihood of death almost tenfold in patients with multimorbidity, as shown by our study.

We can highlight several limitations in our study. The study assessed the outcome (death or recovery) if the patient simultaneously had COVID-19 and one or more diseases, such as diabetes mellitus, atherosclerosis, cardiovascular diseases, and encephalopathy. The selected diseases are both the most common among patients included in the study and the diseases that are most often mentioned as aggravating factors in the severe course of COVID-19, according to the literature. Our review of the published literature found that the most commonly reported chronic conditions in severe SARS-CoV-2 infection were cardiometabolic diseases. Based on previous studies, the main clusters were stroke and hypertension; diabetes and hypertension; CKD and hypertension; and angina and hypertension. At the initial stages of our study, we tried including other pathologies identified in patients (hepatitis, CKD, COPD, bronchial asthma, varicose veins of the lower extremities and gout, benign and malignant neoplasms, etc.) in the model; however, neither as individual predictors nor after inclusion in the model, these pathologies did not show a significant correlation with an increased likelihood of death from COVID-19 [20]. We cannot rule out the possibility that this may be due to the rarity of these conditions among our patients, and therefore, in further studies, it would be interesting to study the effect of vaccination on patients with multimorbid conditions, including these diseases.

The comorbidities we identified as being associated with severe disease and death from COVID-19 were largely similar to those most commonly found in other studies: cardiovascular disease and type 2 diabetes. For example, one study showed that, compared with patients with COVID-19 who did not have a pre-existing history of chronic cardiovascular disease, patients with COVID-19 who have either hypertension or CVD had a risk of developing severe disease of approximately 3–4 times higher [21]. A meta-analysis conducted to examine the putative association between pre-existing diabetes and the severity of COVID-19 found that patients with diabetes had a significantly increased risk (OR: 2.61; 95% CI: 2.05, 3.33) of developing severe COVID-19 compared to patients without diabetes. [16]. This finding is consistent with a previous retrospective study showing that pre-existing diabetes (OR: 3.0; 95% CI: 1.4, 6.3) was a predictor of SARS-related death [22].

Also, it should be noted that it is unvaccinated patients who are most often hospitalized with a new coronavirus infection—more than 85% of patients did not have a single dose of the vaccine in their history, which may lead to some bias in the analysis. Since our study covered the period from 2020 to 2022 and vaccination against coronavirus infection has been carried out mainly since 2021, the bulk of patients have not been vaccinated since 2020. Recommendations on revaccination after a two time primary series of vaccinations came out only on 22 December 2021, in the temporary methodological recommendations of the Ministry of Health of the Russian Federation, “Procedure for vaccination against the new coronavirus infection (COVID-19).” Recommendations for revaccination contain the following provisions: if the level of the immune layer is less than 80–85%—vaccination 6 months after the last dose of the primary vaccination vaccine or the previous revaccination (the number of administered doses is determined by the official instructions for the drug). At the same time, the temporary COVID-19 hospital where the retrospective collection of material for our study was carried out completed the admission of patients in March 2022 (the date of admission of the last patient was 6 March 2022) and was subsequently closed. In this regard, in our study, there were only a few patients who had a history of revaccination. At the same time, it is important to evaluate in further studies the level of protection for patients with multimorbidity after three or more doses of the vaccine against a new coronavirus infection (two vaccinations and revaccinations). There is also great interest in researching the comparative effectiveness of various vaccines in patients with multimorbid conditions. Our study assessed the protective effectiveness of only one vaccine: Sputnik V since the vast majority of patients were vaccinated with two doses of this vaccine (8.4% of patients participating in the study), but it should be noted that a significant proportion of patients are currently immunized in the Russian Federation with other vaccines; therefore, the results of our study can only be applied to patients vaccinated with Sputnik V (Gam-Covid-Vac) since several other vaccines differ in type and mechanism of action.

“Gam-COVID-Vac” is a combined vector vaccine consisting of two components: a recombinant adenoviral vector based on human adenovirus serotype 26, carrying the SARS-CoV-2 S protein gene (component I), and a recombinant adenoviral vector based on human adenovirus 5 serotype, carrying the SARS-CoV-2 S protein gene (component II). The Sputnik Light vaccine is similar in composition to the first component of Gam-COVID-Vac, so we can expect it to be less effective for patients. It should also be noted that the “Salnavak” vaccine has now been released in the form of a nasal spray, completely similar to the “Gam-COVID-Vac” vaccine. However, a more convenient method of administration may increase the population’s adherence to vaccination.

The peptide antigen-based vaccine “EpiVacCorona” included in our study is a chemically synthesized peptide antigen of the S protein of the SARS-CoV-2 virus, conjugated to a carrier protein and adsorbed on an aluminum-containing adjuvant, but in practice, it was shown to be less effective in preventing deaths in patients. The whole virion-inactivated vaccine “CoviVac” is a purified concentrated suspension of the coronavirus SARS-CoV-2 strain “AYDAR-1”, in accordance with the instructions, it showed good efficacy in clinical studies, but it is inferior to “Gam-COVID-Vac” and more often causes side effects after vaccination.

It should be noted here that during different periods of observation, the dominance of certain viral variants was recorded, which could affect both the severity of the disease in patients and the effectiveness of the vaccine [23]. It is therefore promising to study the effectiveness of COVID-19 vaccines in reducing the incidence of infection, hospitalization/severity, and mortality, depending on the strain identified in a patient with multimorbidity, and additional studies may be conducted to test the effectiveness of these vaccines against new emerging variants. Also promising for study is the subunit recombinant vaccine “Convasel”, which is a recombinant nucleocapsid protein (N-protein) of the SARS-CoV-2 virus that remains identical for all strains of the virus.

At the same time, a distinctive feature of our study is an attempt to assess the impact of multimorbidity on the course of COVID-19, in contrast to most studies in which concomitant diseases are taken into account separately at the comorbidity level. However, our results are consistent with the results of a study in which a multivariate analysis was conducted, where among patients hospitalized with COVID-19, the prevalence of multimorbidity (two or more conditions) was more than twice as high in patients with severe SARS-CoV-2 infection (25.3%) compared to patients without severe SARS-CoV-2 infection (11.6%), with the most common clusters found in patients with both stroke and hypertension (79% of stroke survivors had hypertension), diabetes and hypertension (72% of patients with diabetes), and angina and hypertension (67% of patients with angina) [20]. The overall combined effect in patients with multimorbidity (two or more conditions) was associated with approximately 2.6 times the risk of severe SARS-CoV-2 infection compared with patients without multimorbidity (crude OR 2.56 [2.32–2.89]).

In summary, given the high risk of severe disease and high mortality rates among patients infected with SARS-CoV-2 who have multimorbidity, and taking into consideration that organ dysfunction correlates with high mortality rates, treating physicians and other healthcare providers should closely monitor these vulnerable patients. To date, the issue of multimorbidity has not been sufficiently studied by health services or research, which tends to focus on individual diseases or problems. Evidence on the effectiveness of drugs in these patient groups is limited. More trials of treatment and service strategies designed to manage people with clusters of multiple conditions are needed [5,24]. In this regard, our study of the effect of vaccination against a new coronavirus infection in patients with multimorbid pathology on the course and outcome of COVID-19 disease is very important. Persons with a history of multimorbidity may experience potential organ dysfunction, which is further exacerbated by the local and systemic effects associated with SARS-CoV-2 infection. Several studies have examined possible pathophysiological mechanisms linking this infection to severe COVID-19 syndrome, including increased inflammation, decreased immune response, heart failure, kidney failure, and possibly multisystem organ failure and death [20]. For example, one of the features of the new SARS-CoV-2 infection is its ability to induce a systemic hyperinflammatory response accompanied by the instability of atherosclerotic plaques [25]. In turn, COVID-19 can have a negative effect on the endothelium and cardiomyocytes, causing blood clotting and the secretion of pro-inflammatory cytokines and, thus, exacerbating the development of atherosclerosis. The transformation of a stable plaque into an unstable one plays an important role in the pathogenesis of complications of atherosclerosis and can be triggered by COVID-19. However, we would like to point out that further research is needed to understand the underlying biological mechanisms associated with multimorbidity complexes in patients with novel coronavirus infection, the increased risk of severe and fatal COVID-19, and the presence of vaccination, in particular the impact on the immune system. It would also be interesting to see the effect of vaccination on long-term consequences and the post-COVID-19 syndrome.

## 5. Conclusions

In our study, we present a way to simultaneously assess two global health problems: the rapidly growing prevalence of the population who have had SARS-CoV-2 and the growing number of people living with multimorbidity around the world. It has now been shown that these two conditions can potentiate severe disease and adverse outcomes when interacting with each other. This interaction, as well as the effect of specific SARS-CoV-2 vaccinations in these patients, is expected to have a non-linear relationship but represents important clinical and public health implications that are missed when focusing on individual diseases.

Our study confirmed that patients with multimorbidity are at increased risk of death from a new coronavirus infection. Thus, in the first model, when assessing the outcome of a new coronavirus infection depending on the presence of a multimorbid condition in patients, including a combination of pathologies such as cardiovascular disease, atherosclerosis, and type 2 diabetes mellitus, the chances of death increased in the presence of two of these pathologies by 4.6 times and in the presence of three of the listed pathologies by 9.4 times. When analyzing the second model, we found that in patients with a combination of several pathologies, the chances of death increased by 1.6 times in the presence of diabetes mellitus, by 4.4 times in the presence of atherosclerosis, and by 2.3 times in the presence of established encephalopathy. Considering this, vaccination against coronavirus infection in patients with two or more chronic diseases is an important preventive measure that reduces the risk of a severe course with an unfavorable outcome by more than three times. In connection with the above, we can conclude that it is vaccination against COVID-19 that will allow people with multimorbidity to avoid severe complications when developing a new coronavirus infection (COVID-19), even in the case of severe disease.

In conclusion, it should be noted that although, on the one hand, chronic diseases contribute to increased susceptibility to infectious diseases that can be prevented by vaccination, on the other hand, age-related concomitant diseases can worsen the immune response to vaccination, reducing their clinical effectiveness [25]. Thus, there is a need for further research into the complex interactions between comorbidities and the immune system to develop more effective vaccines and optimize recommendations for people with multimorbidity.

## Figures and Tables

**Figure 1 vaccines-11-01696-f001:**
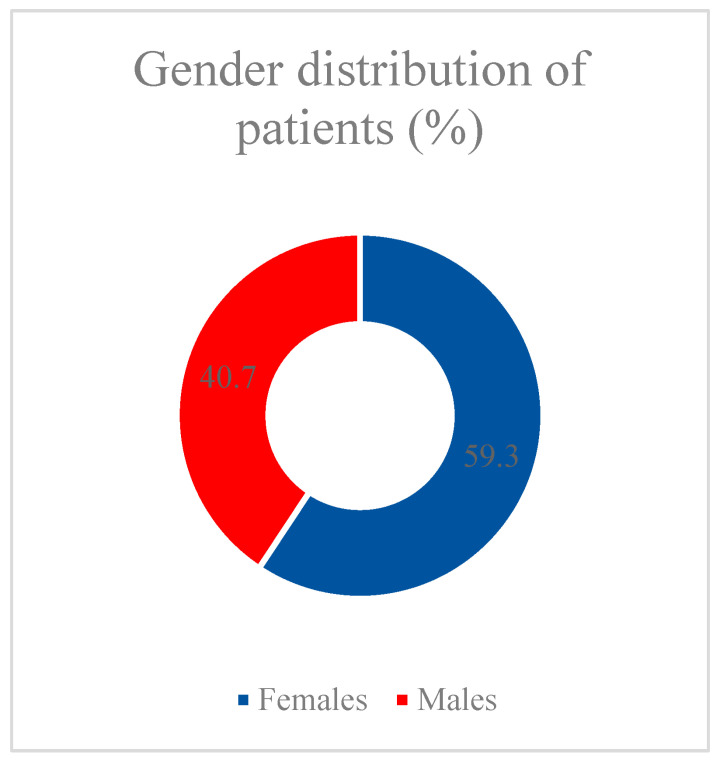
Gender distribution of patients.

**Figure 2 vaccines-11-01696-f002:**
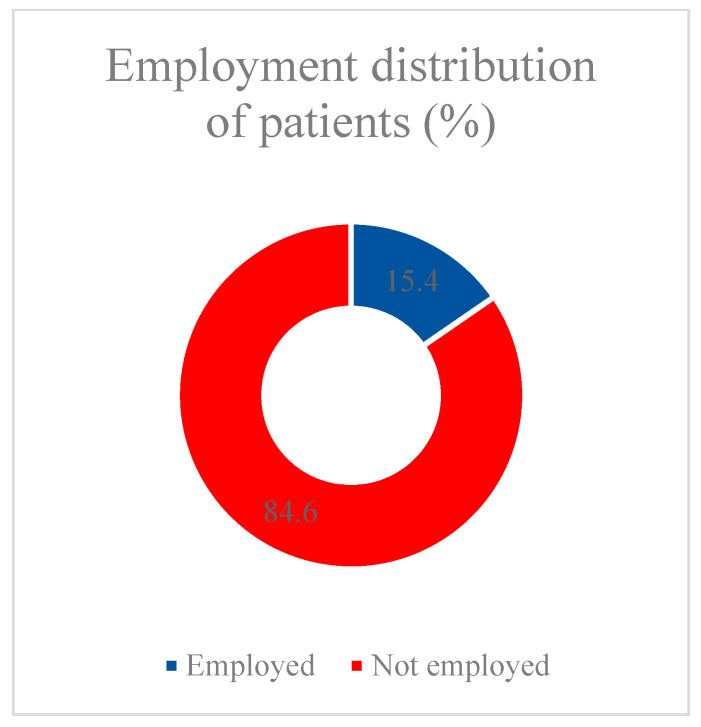
Employment distribution of patients.

**Figure 3 vaccines-11-01696-f003:**
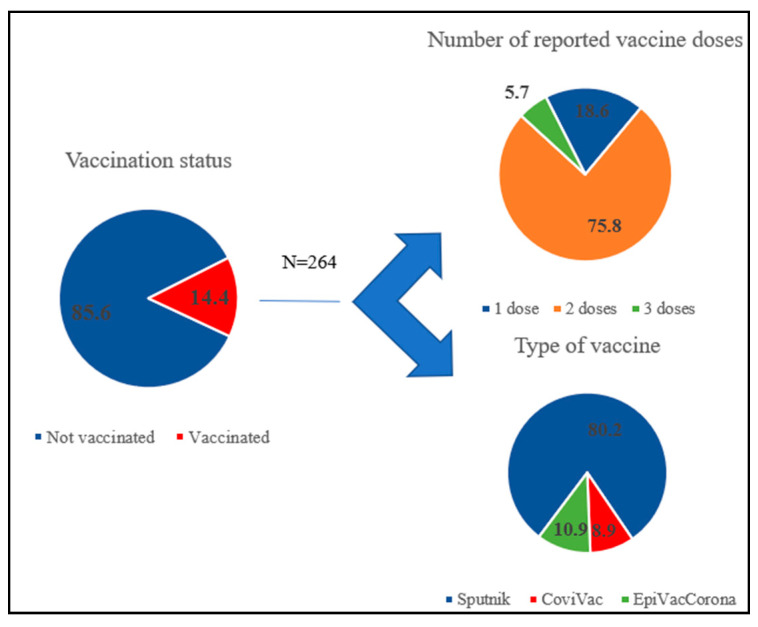
Vaccination history of patients.

**Figure 4 vaccines-11-01696-f004:**
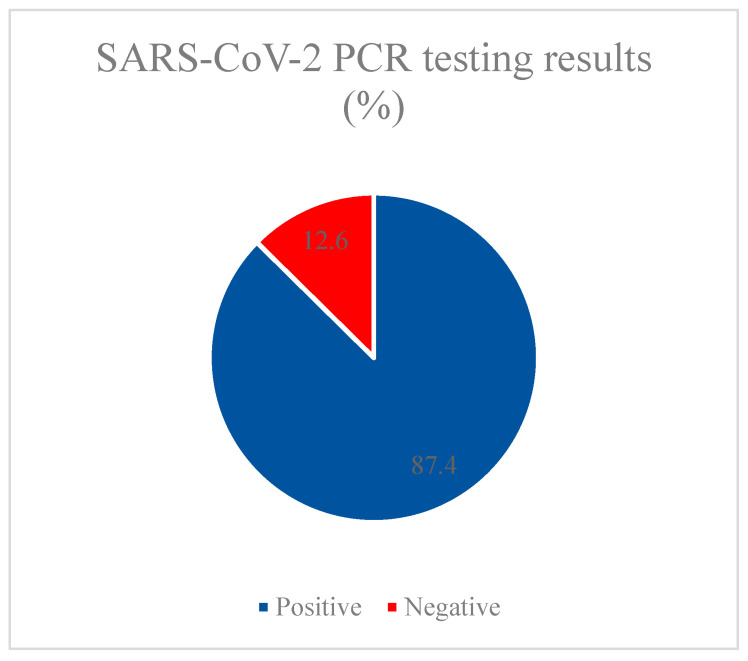
SARS-CoV-2 PCR testing results.

**Figure 5 vaccines-11-01696-f005:**
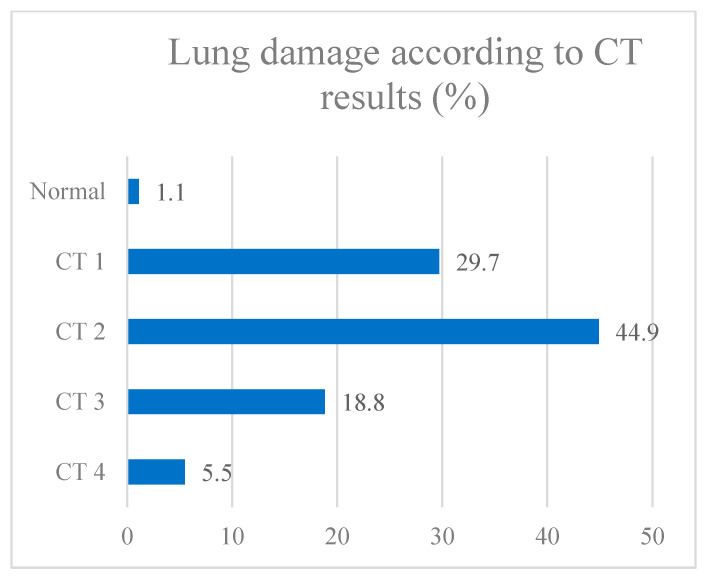
Lung damage according to CT results.

**Figure 6 vaccines-11-01696-f006:**
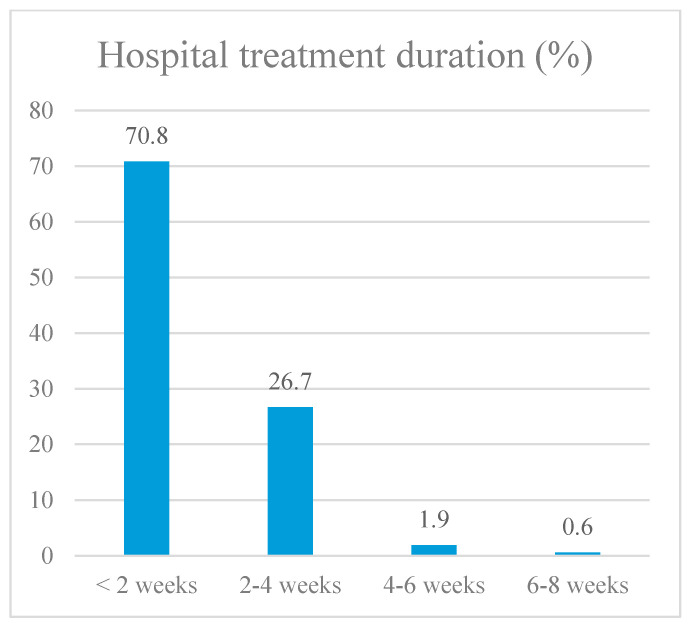
Hospital treatment duration.

**Figure 7 vaccines-11-01696-f007:**
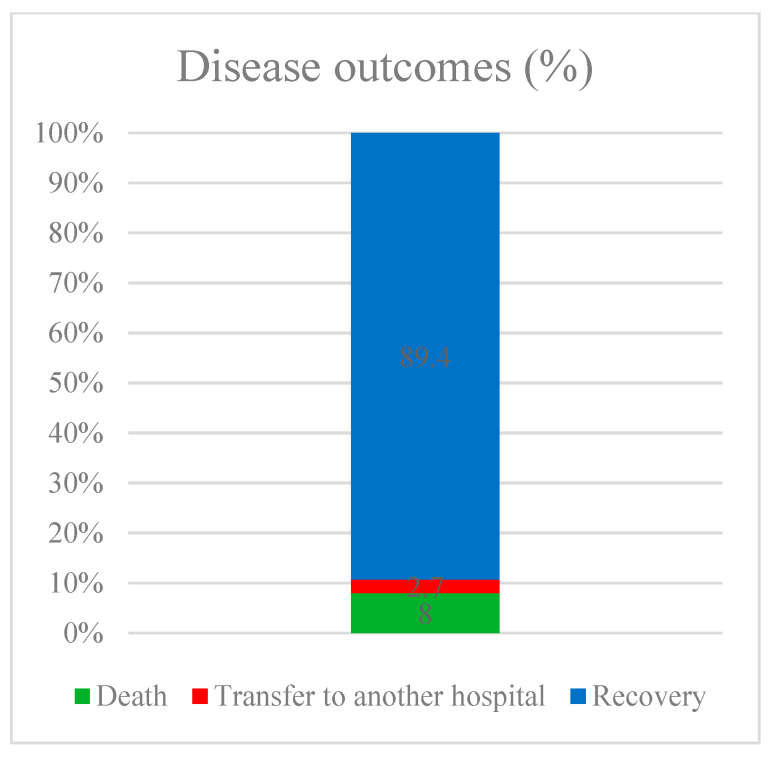
Disease outcomes.

**Figure 8 vaccines-11-01696-f008:**
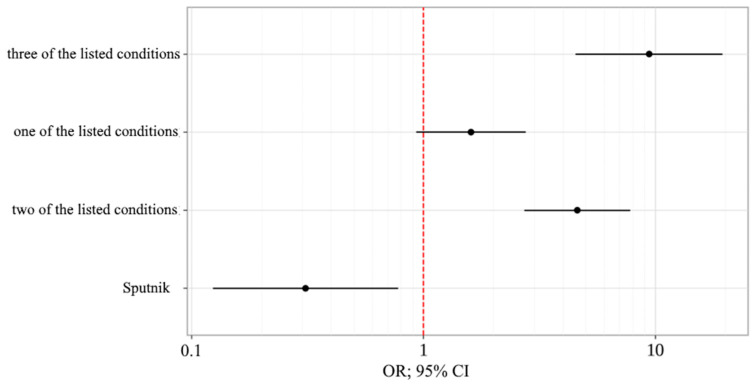
Odds ratio estimates for studied outcome-coding predictors with a 95% CI (Model 1).

**Figure 9 vaccines-11-01696-f009:**
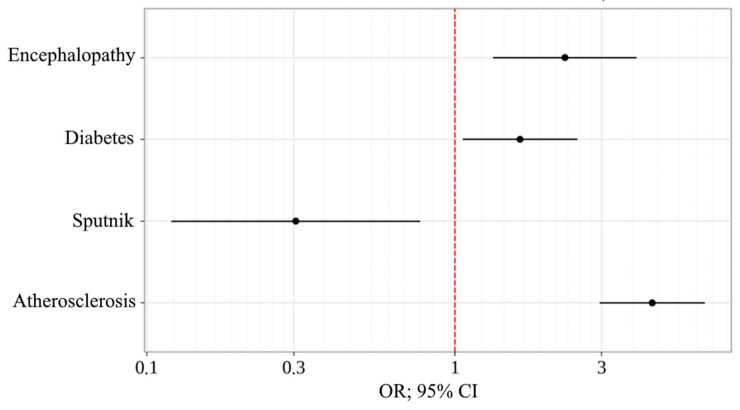
Odds ratio estimates for studied outcome-coding predictors with a 95% CI (Model 2).

**Table 1 vaccines-11-01696-t001:** Burden of COVID-19 in the Russian Federation, 2020–2022.

	Incidence (per 100,000 Persons)	Total Cases	Cases per Day (Average)	Total Deaths (Cumulative Sum)	Deaths per Day (Average)	Lethality, %
2020	682.85	1,002,977	10,709	16,908	193	1.40%
2021	4198.15	6,166,268	20,111	156,602	690	2.44%
2022 ^1^	7489.49	11,000,598	49,358	321,717	722	2.93%

^1^ Available data up to February 2022.

**Table 2 vaccines-11-01696-t002:** Analysis of adherence to vaccination depends on the employment of patients.

Indicator	Category	Employment		*p*
		Employed	Not Employed	
Vaccination	Not vaccinated	228 (80.6%)	1340 (86.5%)	0.009 *
	Vaccinated	55 (19.4%)	209 (13.5%)	

*—differences in indicators are statistically significant (*p* < 0.05).

**Table 3 vaccines-11-01696-t003:** Analysis of the hospital stay duration depends on the patient’s vaccination history, taking into account the number of doses and type of vaccines.

Indicator	Categories	Duration of Stay			*p*
		Me	Q₁–Q₃	n	
Vaccination	Not vaccinated	12	9–16	1548	<0.001 * p_Sputnik–not vaccinated_ < 0.001
	Sputnik	10	7–13	155	
	CoviVac	10	9–13	19	
	EpiVac	12	10–15	26	
	Sputnik Light	9	8–10	2	
	Sputnik 1 dose	11	8–14	40	
	EpiVac 1 dose	42	28–42	3	
	CoviVac 1 dose	10	8–12	4	
	Sputnik + Light	10	9–11	15	

*—differences in indicators are statistically significant (*p* < 0.05).

**Table 4 vaccines-11-01696-t004:** Relationship between predictors of the model and the probability of death or recovery (Model 1).

Predictors	Unadjusted		Adjusted	
	COR; 95% CI	*p*	AOR; 95% CI	*p*
CVD + Atherosclerosis + Diabetes: one of the listed conditions	1.624; 0.944–2.795	0.080	1.602; 0.931–2.759	0.089
CVD + Atherosclerosis + Diabetes: two of the listed conditions	4.518; 2.672–7.637	<0.001 *	4.604; 2.721–7.791	<0.001 *
CVD + Atherosclerosis + Diabetes: three of the listed conditions	9.091; 4.397–18.784	<0.001 *	9.402; 4.527–19.511	<0.001 *
Vaccination: Sputnik	0.353; 0.142–0.876	0.025 *	0.310; 0.123–0.777	0.013 *

*—the influence of the predictor is statistically significant (*p* < 0.05).

**Table 5 vaccines-11-01696-t005:** Relationship between predictors of the model and the probability of death or recovery (Model 2).

Predictors	Unadjusted		Adjusted	
	COR; 95% CI	*p*	AOR; 95% CI	*p*
Vaccination: Sputnik	0.353; 0.142–0.876	0.025 *	0.305; 0.120–0.773	0.012 *
Presence of diabetes	1.811; 1.206–2.721	0.004 *	1.629; 1.061–2.504	0.026 *
Presence of atherosclerosis	5.374; 3.743–7.714	<0.001 *	4.378; 2.956–6.482	<0.001 *
Presence of encephalopathy	4.838; 2.989–7.830	<0.001 *	2.279; 1.331–3.900	0.003 *

*—the influence of the predictor is statistically significant (*p* < 0.05).

## Data Availability

The data presented in this study are available on request from the corresponding author. The data are not publicly available due to the published data including only part of the work.

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
