# Peer review of "Impact of Vaccination on the Course and Outcome of COVID-19 in Patients with Multimorbidity"

_vaccines, 2023, doi:10.3390/vaccines11111696_

Round 1
Reviewer 1 Report
Comments and Suggestions for Authors
In the present work, the authors present in the title a study on the effect of the COVID vaccine on the survival of patients with multimorbidities. The work is interesting and is little reported in the literature. The body of the manuscript presents a large amount of data that bear little relation to what is described in the title and the main objective of the study. Authors should stick to the described objective, which is what the title of the manuscript describes and generates interest. When the manuscript discusses multimorbidity, the data is presented in an unclear way, and should be presented in a more clarified way, comparing mortality in patients with multimorbidity between vaccinated and unvaccinated patients.
Author Response
Dear Sir/Madam! Thank very much you for your review. Please see the attachment

Reviewer 2 Report
Comments and Suggestions for Authors
Comments
1. Authors need to update the introduction and Aim in the abstract.
2. Abstract should consist of Background, Aim, methods, results and conclusion. Kindly update it.
3. I think right now COVID-19 is not an urgent problem in the present scenario.
4. Authors need to add the prevalence of COVID-19 in Moscow population.
5. Also, authors need to update the frequencies of vaccinations applied population of Moscow?
6. Does this study approve the ethical grant?
7. In Table 1, authors need to add the percentages also.
8. How were the figures (figure 1-9) was created and authors need to add the software details for all the tables.
9. There are many countries who were comes under multimorbidity. It will be better if the authors describe the top 5 countries with multimorbidity.
10. The authors need to compare the current study results with the global wide studied studies.
11. Describe limitations and strengths of this study.
12. Rewrite the conclusion and recommend the future studies
Author Response

(The authors gave the same response as above.)

Reviewer 3 Report
Comments and Suggestions for Authors
ID: vaccines-2650667
Impact of Vaccination on the Course of COVID-19 in Patients with a Negative Medical History. by Lomonosov K, et al.
To the Authors:
General comments:
The authors investigated the effects of Russian vaccines for SARS-CoV-2 in patients with multimorbidity. They concluded that the presence of two doses of Sputnik V increased the likelihood of recovery in patients with multimorbidity by more than three times. It was considered that the topic was important, and the results included novelty; however, several points should be addressed to improve the manuscript.
Specific comments:
1. Please explain the reasons why the authors chose the diseases that consist of Model 1 and Model 2 of the multimorbidity in the patients.
2. Please add the Figure legends in Figure 1-9. Also, stages of CT1-4 should be defined in the manuscript (lines 158-165).
3. Please confirm that this study is ethically approved.
4. The present title may not be suitable for the context in this paper. ‘Patients with negative history’ would be changed to ‘patients with multimorbidity’.
5. The authors should discuss the possible mechanism of the various vaccination effects on the patients with multimorbidity.
Author Response

(The authors gave the same response as above.)

Round 2
Reviewer 1 Report
Comments and Suggestions for Authors
The authors have modified the manuscript according to the suggestions made in the previous review.
The results about mortality could be presented more clearly from my point of view. It is not clear whether the vaccinated group corresponds to patients with multimorbidity or patients in general.
Author Response
Dear Sir/Madam,
Thank you once again for your assessment. Your notes and questions have been invaluable in preparing this manuscript. The data in the vaccinated group represents patients with multimorbidity. We have included clarifications in the new version in the manuscript, please find them outlined in light blue, particularily at lines 23-24, 311, 455-457
Kind regards,
The authors
Reviewer 2 Report
Comments and Suggestions for Authors
Accept
Author Response
Dear Sir/Madam,
Thank you once again for your assessment. Your notes and questions have been invaluable in preparing this manuscript.
Kind regards,
The authors
Reviewer 3 Report
Comments and Suggestions for Authors
ID: vaccines-2650667
Impact of Vaccination on the Course of COVID-19 in Patients with a Negative Medical History. by Lomonosov K, et al.
To the Authors:
General comments:
It is considered that the authors successfully revised the manuscript according to the comments. It is necessary to mention the information regarding the differences among Sputnik, EpiVac, CoviVac and Sputnik+Light in the introduction or discussion sections.
Author Response
Dear Sir/Madam,
Thank you once again for your assessment. Your notes and questions have been invaluable in preparing this manuscript. We absolutely agree that the information on the differences between vaccines should have been presented and we have updated the manuscrip accordingly. Kindly find the updated parts outlined in light blue, particularly lines 368-386, 393-395
Kind regards,
The authors